# Edible Quality Analysis of Different Areca Nuts: Compositions, Texture Characteristics and Flavor Release Behaviors

**DOI:** 10.3390/foods12091749

**Published:** 2023-04-23

**Authors:** Bowen Yang, Haiming Chen, Weijun Chen, Wenxue Chen, Qiuping Zhong, Ming Zhang, Jianfei Pei

**Affiliations:** 1Hainan University-HSF/LWL Collaborative Innovation Laboratory, School of Food Science and Engineering, Hainan University, Haikou 570228, China; 18224463713@163.com (B.Y.); 992984@hainanu.edu.cn (H.C.); chenwj@hainanu.edu.cn (W.C.); chwx@hainanu.edu.cn (W.C.); hainufood88@163.com (Q.Z.); zhangming-1223@163.com (M.Z.); 2Huachuang Institute of Areca Research-Hainan, 88 People Road, Haikou 570208, China

**Keywords:** areca nut, compositions, texture characteristics, flavor release behaviors

## Abstract

The areca nut is one of the most important cash crops in the tropics and has substantial economic value. However, the research information about the edible quality of different areca nuts is still insufficient. This study compared the composition, texture characteristics and flavor release behaviors of four different areca nuts (AN1, AN2, AN3 and AN4) and two commercially dried areca nuts (CAN1 and CAN2). Results showed that AN1 had higher soluble fiber and lower lignin, which was the basis of its lower hardness. Meanwhile, the total soluble solid (TSS) of AN1 was the highest, which indicated that AN1 had a moister and more succulent mouthfeel. After the drying process, the lignification degree of AN1 was the lowest. Through textural analyses, the hardness of AN1 was relatively low compared to the other dried areca nuts. AN1, CAN1 and CAN2 had higher alkaline pectin content and viscosity, and better flavor retention, which indicated better edible quality. The present study revealed the differences of various areca nuts and provided vital information to further advance the study of areca nuts.

## 1. Introduction

The areca nut is a small, fibrous seed endosperm grown from the Areca catechu L. palm [1]. Being a customary and traditional commodity, areca nut consumption is socially acceptable among all sections of society [2]. Globally, the areca nut is the fourth most commonly used psychoactive substance after tobacco, alcohol and caffeine [3,4]. Areca nuts can help with resistance to bacteria, removal of parasites and facilitation of digestion [5]. Estimated 600 million people consume areca nuts, predominantly in the southern parts of Asia such as China (Hainan and Taiwan), Vietnam, Indonesia, Myanmar and the Pacific regions [6].

Areca nuts are used at different stages of maturity as the whole nut or in thin slices, in a natural state or after processing in many forms. There are three main ways to chew areca nuts: fresh areca nuts (Southeast Asia), dried kernel (India) and dried areca nut pericarp (China). Fresh areca nuts are typically chewed with slaked lime and Laotian leaves or grated tobacco [7]. In China, dried areca nut is typically packaged into commercial products by boiling, adding flavor, stewing until it is fragrant, adding brine and drying. The flavor of the areca nut is highly related to consumer preference. The usual approach of flavor improvement is by marinating the areca nut with some additives (such as sweeteners). During the process of chewing the areca nut, those additives release a unique taste and smell. Therefore, flavor release behaviors are the key parameters in evaluating the edible quality of the areca nut.

Previously, it has been reported that chewing areca nuts may cause oxidative damage in the oral mucosa [8,9]. Research revealed the high-risk factors of oral cancer in South Asia that were related to the consumption of areca nuts [10]. The process of chewing the areca nut renders more mechanical friction between oral mucosa and the crude fiber, resulting in a higher possibility of chronic mucosa lesions and thereby the development of cancer [11]. In order to reduce the risks associated with areca nut consumption, areca nuts with flexible and thin fibers and abundant soluble solids are the preferred choice for consumers.

Therefore, an edible quality analysis of different areca nuts is particularly important to minimize the damage and improve the market status. In this study, the composition, texture characteristics and flavor release behaviors of different areca nuts were compared. The research information will promote harmless areca nut consumption and market development.

## 2. Materials and Methods

### 2.1. Materials

The areca nuts were from China (Hainan, AN1), Vietnam (AN2), Indonesia (AN3) and Myanmar (AN4). All samples were purchased at local wholesale markets (Haikou, Hainan, China) and transported under refrigeration at 4 °C throughout. Samples were selected for their uniformity of size and color and the absence of mechanical damage or diseased material. Areca nuts were randomly selected from each producing area, and their aspect ratios were measured by vernier calipers with a precision of 0.02 mm. Testing for each sample was repeated 16 times. Some areca nuts were cut into small pieces and pressed. After pulping, the puree was passed through filter cloths of 150 mesh (100 μm) and stored at 4 °C. The fresh areca nuts were placed in an oven at 50 °C for 72 h to prepare dried areca nuts. The commercially dried areca nuts kouweiwang (CAN1) and hechengtianxia (CAN2) were bought from a local market. 

### 2.2. Analysis of Fiber from the Areca Nut Samples

#### 2.2.1. Total Fiber

The areca nut samples were dried and crushed. The powder of the areca nut was added to a nitric acid–ethanol mixture (1:4), which was then precipitated for 5 h at 95 °C, followed by filtration on filter paper. The residue was washed twice with 10 mL of nitric acid–ethanol mixture, hot water and 95% ethanol in sequence and repeated once. Finally, the residue was oven-dried at 55 °C to a constant weight and then stored at 4 °C [12]. Testing for each sample was repeated five times.

#### 2.2.2. Lignin Content

The content of acid-insoluble lignin was determined using the concentrated sulfuric acid method. Briefly, 2 g areca nut powder and 30 mL 72% concentrated sulfuric acid were added to react at 25 °C for 4 h. Then, 345 mL distilled water was added, and the whole system was boiled for 4 h. Finally, the residue was obtained by filtration and oven-dried at 55 °C to a constant weight [13]. Testing for each sample was repeated five times.

#### 2.2.3. The Soluble Fiber

The areca nut powder was degreased with 4 volumes of petroleum ether for 3 h and centrifuged at 6000 rpm for 5 min at 4 °C on a centrifuge (H1850R, Hunan Xiangyi Laboratory Instrument Development Co., Ltd., Changsha, China) to remove the supernatant fluid, and then mixed with 0.1 M hydrochloric acid at a ratio of 1:20 (*w*/*v*) at 80 °C for 10 h. The supernatants were isolated via 4 volumes of 95% ethanol precipitation. Three replicates were performed for each sample [14].

#### 2.2.4. Alkaline Pectin

The areca nut fiber was added into 0.5 mol/L NaOH solution at a ratio of 1:15, then stirred and extracted for 3 h at 80 °C. Secondly, the mixture was filtrated and then added to isopropanol for 12 h at 4 °C, with alcohol precipitation at 4 °C for 12 h. The alcohol-insoluble substance was centrifuged and washed by isopropanol three times. Finally, the samples were redissolved in deionized water and then freeze-dried [15]. Different pectin solutions (40.0 g/L) were prepared in deionized water using magnetic stirring for 3 h at 25 °C. The rheological properties of different pectin solutions were determined with a rheometer (RST-CC, Brookfield Inc, Middleboro, MA, USA) using concentric cylinder geometry (length = 40 mm, diameter = 26.66 mm, gap width = 4000 μm) at 25 °C. The viscosity standard fluid was used for calibration at 25 °C. Flow curves were obtained for a shear rate sweep between 0.1 and 100 s^−1^ (the increase rate was fixed to 0.278 s^−1^, and the experimental data were taken every 6 s), and the viscosity of different pectin solutions was compared at 40 s^−1^ shear rate [16]. Testing for each sample was repeated three times.

#### 2.2.5. Morphological Characterization and the Moisture Content of the Areca Nut Fiber

The microstructure of the areca nut fiber was observed using a fluorescence microscope (AE31, MOTIC China Group Co., Xiamen, China) [17]. The moisture content of the areca nut fiber was tested using the direct drying method [18]. The initial weight (W1) of the randomly selected areca nut fiber was determined and recorded using an electronic weighing balance (PL3002, METTLER TOLEDO Group, Zurich, Switzerland) with a sensitivity of 0.01 g. Then, the samples were put in an electric heating dryer (DHG-9140, Shanghai Yiheng Scientific Instruments Co., Ltd., Shanghai, China) at 105 °C and dried to constant weight (W2); prior to each weighing, the sample was cooled in a desiccator. The moisture content was determined by the relation (W1 − W2)/W1. Testing on each sample was repeated five times.

### 2.3. Textural Analysis

#### 2.3.1. Compression Test of Fresh Areca Nuts

A texture analyzer (TA XT plus C, Stable Micro Systems, Godalming, UK) with P50 probe and a 50 kg load cell was used in this study. The areca nuts were cut in half and placed on the testing platform in parallel for a compression test (TPA mode), which was conducted with a pre-test speed of 5 mm/s, test speed of 1 mm/s, post-test speed of 5 mm/s, trigger force of 5 g and compressed depth of 35% [19]. Testing for each sample was repeated 12 times.

#### 2.3.2. Puncture Test of Fresh Areca Nuts

The puncture probe (P2N) was used for the puncture test: pre-test speed 5 mm/s, test speed 1 mm/s, post-test speed 5 mm/s and trigger force 5 g. The largest force value was recorded as the hardness in a force–time curve. Each sample was punctured by probe to 3 mm [20]. Testing for each sample was repeated 12 times.

#### 2.3.3. Fiber Hardness Test of Dried Areca Nuts

Referring to the methods of Wetchakama et al. [21] (with a few modifications), the fiber texture of the areca nut was determined using a three-point bend method on a texture analyzer equipped with an HDP-3PB probe. A force–time curve was generated from the compression. The largest force value was taken as the measure of hardness. A longitudinal areca nut piece (1 cm × 2 cm) was placed on the two base supports at a distance (span) of 1 cm, with the epidermis facing upward. The test conditions set for the instrument were a pretest speed of 3.00 mm/s, a test speed of 1.00 mm/s, a post-test speed of 3.00 mm/s, a trigger force of 20.0 g and a deformation of 40%. Testing for each sample was repeated 12 times.

### 2.4. Analysis of Pressed Liquid from Areca Nuts

#### 2.4.1. The Total Soluble Solids

The total soluble solids (TSS) of the areca nuts’ pressed liquid were measured using a refractometer (WAY, INESA, Shanghai, China). Testing for each sample was repeated five times.

#### 2.4.2. The Total Phenolics

The total phenolics of the areca nuts’ pressed liquid were analyzed using spectrophotometric determination (the Folin–Phenol method) [22]. A standard curve was prepared using a gallic acid series from 10 to 50 μg/mL for testing. A total of 1.0 mL from each test solution was mixed with 5.0 mL of 10% Folin–Phenol solution, reacted for 5 min, and then 4.0 mL of 7.5% Na_2_CO_3_ solution was added to the reaction at room temperature for 1 h in the dark. The absorbance was measured at 765 nm using a UV-visible spectrophotometer (TU-1810, Beijing Puxi General Instrument Co., Beijing, China). The results were calculated with the standard curve and were expressed as mg of gallic acid equivalents (GAE) per 1 mL of areca nut pressed liquid (mg GAE/mL). Testing for each sample was repeated five times.

#### 2.4.3. The Molecular Weight

The size exclusion chromatogram was used to study molecular weight (MW) distribution profiles of the areca nuts’ pressed liquid. The size-exclusion chromatography analyzer setup consisted of a HPLC system (Shimadzu Corporation, Kyoto, Japan) equipped with a TSK G4000SWXL column. The mobile phase was 0.9% NaCl water solution with 0.5 mL/min of flow rate. Sample runs were calibrated daily. Then, the solutions were filtered through a 0.45 μm PES membrane before analysis, and the injection volume was 80 μL [23].

### 2.5. Flavor Release Test

The dried areca nuts were immersed in brine, which was packed with some ingredients at 60 °C for 60 h (sodium cyclamate as a sweetener and orange oil as a flavoring agent), to prepare edible areca nuts. The homebrew experimental set-up was used for the edible areca nuts’ flavor release test. This method is a modification of previous methods for simulating chewing using a texture analyzer [24,25]. As shown in Figure 1, artificial saliva (2 mL/min) was transported by constant-flow pump to a texture analyzer testing platform. The test conditions set for the instrument were a pretest speed of 10.0 mm/s, a test speed of 2.0 mm/s, a post-test speed of 10.0 mm/s, a trigger force of 20.0 g and a deformation of 65%. The areca nuts were pressed by P50 probe to simulate chewing. After 3 min, samples were collected at 3, 3.5, 4, 5, 7, and 11 min to measure the flavoring ingredients (or samples were collected at 10, 30-, 40-, 50- and 60-times simulated chews). The artificial saliva was emptied after each collection, and the areca nut was turned over constantly during this time. The cyclamate was measured using high-performance liquid chromatography (HPLC) equipped with a variable wavelength, UV–visible detector measuring at 230 nm and a C18 column (2.5 μm, 4.6 mm × 250 mm). All chromatograms were performed at room temperature with the use of a 95:5 (*v*/*v*) mixture of ammonium acetate (0.25%) and methanol, and a flow rate of 1 mL/min [26]. The orange oil (mainly limonene) was measured using gas chromatography equipped with A HP-5MS UI GC column (0.25 μm, 30 m × 0.25 mm) [27]. Helium gas was used as the carrier gas at a flow rate of 0.9 mL/min, and the sample inlet temperature was set at 275 °C. The sample injection volume was 1 μL. The temperature gradient was as follows: 50 °C for 5 min, raised to 90 °C at 5 °C/min, held for 2 min, raised to 96 °C at 2 °C/min and held for 2 min. After the gradient was completed, the post-run was set to proceed at 260 °C for 1 min. The following MS conditions were used: mass range, m/z 45–200. Testing for each sample was repeated five times.

### 2.6. Statistical Analyses

Values were expressed as the mean value ± standard deviation. Statistical analyses were performed using IBM SPSS Statistics 26 software (IBM Corporation, Armonk, NY, USA). The differences between groups were distinguished using one-way ANOVA and Duncan’s multiple range tests, and *p* < 0.05 was considered statistically significant and indicated by different letters. Some data were plotted using Origin2022b software (OriginLab Corporation, Northampton, MA, USA).

## 3. Results

### 3.1. Visual Appearance of Areca Nuts

The appearance of a food product is an important quality that determines its first impression. This is also true for areca nuts as a local specialty and chewable product. As shown in Table 1 and Figure 2, all areca nuts are dark green in color and olive-shaped. AN3 had the highest length/width rate (1.80), and it was the narrowest sample among these areca nuts. The length/width rates of AN1, AN2 and AN4 ranged from 1.63 to 1.69. When cut open, it can be seen that areca nut can be divided into kernels and peels. The peel is mainly composed of cellulose, lignin and hemicellulose, and the kernel is rich in phenols and water, which makes it easy to brown [28]. The grooves of dried areca nuts are clear and delicate, which could meet the processing requirements.

### 3.2. Analysis of Fiber from Areca Nuts

The fiber characteristics are closely related to the taste quality of areca nuts. The morphology of the areca nut fibers is shown in Figure 3. The AN1 fibers were the thinnest and had a certain curvature, showing that the AN1 fiber was relatively more slender and softer among these samples. However, the fibers of AN3 and AN4 were thicker. The flocculent components between the fibers were mostly polysaccharides (such as hemicellulose), which could be what gives areca nuts their moist and succulent taste. There are more flocculent components in AN1 and AN4. This finding is consistent with that of Shi et al. [29] who also found that the crude fiber component is one of the most important factors affecting the taste quality of Brassica napus.

### 3.3. Composition of Fresh Areca Nuts

The compositions of the areca nuts were summarized in Table 2. There was a positive correlation between the content of TSS and the dregs solubility of the areca nut. The TSS in AN1 was as high as 4.94%, which was higher than that of AN2, AN3 and AN4. Soluble fiber mainly consists of pectin polysaccharide, which directly determines the areca nut’s taste [30]. Hu et al. [31] also found that soluble polysaccharide is the key factor determining the taste quality of fruit. It can be seen from Table 2 that AN1 had the highest soluble fiber content (3.00 ± 0.24%), while AN2 and AN4 showed the second highest content. There is no significant difference between the areca nuts in fiber moisture content.

There are many polyphenol components in areca nuts. These polyphenol components are mostly tannins, which have a strong astringency [32]. Polyphenol components should be removed with high temperature, alkali foaming and other processes [33]. As shown in Table 2, AN1 had the lowest total polyphenols (2.76 mg GAE/mL). As such, it is less difficult to reduce the polyphenol content of AN1, which decreases the cost of downstream processing.

Lignin, as a cross-linked phenolic polymer, is the important intrinsic component that bonds tightly to cellulose, providing the rigidity and hydrophobic property of the fruit [34]. Therefore, the lignin content was positively correlated with the hardness of the areca nuts. The greater hardness of the areca nut fiber will lead to a lower mouthfeel and even cause oral damage [11]. As shown in Table 2 (express as mg of lignin per 1 g dry weight), the lignin content of AN1 was 24.81%, which was lower than AN2 (25.77%), AN3 (28.31%), and AN4 (26.01%). 

### 3.4. Textural Analysis of Fresh Areca Nuts

The analysis of the texture data is primarily based on the relationship between distance, force and time. The texture profile analysis (TPA) test, which is based on the imitation of mastication, or the chewing process, is performed with double-compression cycles. The different probes of the texture analyzer are shown in Figure 4. The hardness is determined by the maximum force applied under the experimental parameters. As the probe is pressed down, the greater the counter-force generated by the areca nut fibers, the progressively higher the hardness value measured by the device. Springiness and chewiness were calculated based on the force/distance data collected during the two-cycle compression test. Springiness represents the height ratio before and after compression, and chewiness represents the energy from chewing to swallowing of the solid sample [35].

The texture properties of areca nuts in various chewing stages are different. The puncture test can reflect the hardness of the first chewing stage. The food texture properties of the subsequent stage could be indicated in the compression test. As shown in Table 3, AN1 had the lowest hardness both in the puncture test and the compression test, which matches the results in the fiber analysis.

### 3.5. MW Distribution

Polysaccharides can be leached from areca nuts during the chewing process. These polysaccharides with large MW can improve the succulent taste [36]. The size exclusion chromatogram using a HPLC system was used to determine MW distribution profiles of the areca nuts’ leachability, and the results are shown in Figure 5. Among these samples, AN1 had a unique macromolecule peak (101,647 Da), which can provide a succulent mouthfeel. There was no significant difference in MW distribution profiles between AN2, AN3 and AN4.

### 3.6. Fiber Composition of Dried Areca Nuts

Dried areca nuts are one of the most commonly eaten forms of the nuts. Generally, the drying process leads to an increase in the lignification degree of the areca nuts. Comparing Table 2 and Table 4, it could be seen that the drying process resulted in a decrease in soluble fiber content and an increase in lignin [37]. As shown in Table 4, the soluble fiber of AN1 was the highest, and the lignin of AN1 was the lowest; meanwhile, there was no significant difference in soluble fiber between AN2, AN3 and AN4.

### 3.7. Textural Analysis of Dried Areca Nut Fiber

For the texture of dried areca nut fiber measurements, the sample was cut into small pieces to produce a flatter sample and to minimize the detrimental effect of the areca nut’s internal cavity. The hardness was determined by the maximum force applied under the experimental parameters. The textural parameters of homebrew and commercially dried areca nuts were summarized in Table 5. CAN1 and CAN2 had the lowest hardness, chewiness and springiness. The hardness closest to CAN1 and CAN2 was AN1, which has the lowest lignin content (Table 4).

### 3.8. Alkaline Pectin

The consumption of dried areca nuts is commonly accompanied with alkaline brine. Therefore, the content and viscosity of alkaline pectin in dried areca nuts can directly influence the succulent mouthfeel [38]. Furthermore, the alkaline pectin can reduce the friction between the oral mucosa and areca nuts, thereby reducing the potential health risk. Pectin solution behaves as non-ideal, i.e., non-Newtonian liquids, in which the viscosity decreases with the increase of shear rate, and the fluid is shear-thinning [16]. As shown in Table 6, the content of alkaline pectin was the highest among the three samples of AN1, CAN1 and CAN2. Meanwhile, the alkaline pectin viscosities (at the same concentration level) of these three samples were also the highest. It indicated that AN1, CAN1 and CAN2 had better succulent mouthfeels. A possible explanation for this might be that the different glycosidic bond and the branching structure of pectin result in different viscosities [39].

### 3.9. Flavor Release of Dried Areca Nuts

The texture analyzer is an instrument for measuring various mechanical properties of products based on simulated chewing; however, there are few reports of using a texture analyzer to simulate chewing. Liu et al. [24] simulated the oral chewing of rice based on a texture analysis. Kim et al. [25] used a texture analysis to simulate the chewing of rice flour to study the effect on the structure, exploring the release behavior of betel nuts based on texture analysis. To better simulate chewing, experimental time, amount of artificial saliva and number of simulated chews were determined based on the observation of areca nut chewing behavior and related literature [40]. Because of the different solubility of the substance, different methods of simulated chewing and data processing were used.

The flavor release behaviors, which were represented by the flavor ingredient’s variation over chewing times, were important properties affecting the areca nuts’ quality. The flavor release behaviors of dried areca nuts are shown in Figure 6. AN1 had the highest initial sodium cyclamate contents. This result implied that it was easier to obtain flavor ingredients in AN1, under the same conditions. As the number of chewing times increases, the sodium cyclamate content downtrend of AN1 is always higher than AN2, AN3 and AN4. As shown in Figure 6b, both samples had gradually decreased orange oil content. The curve slopes of AN1 were lower than AN2, AN3 and AN4, indicating that the aroma ingredients of AN1 and the commercial products were released slowly. As shown in Figure 6, due to the large amount of sugar dissolution in the brine of the commercial areca nut, the release of soluble solid gradually decreases with the increase of chewing time. There was a more rapid release of soluble solids from CAN2 compared to CAN1. However, there was no significant difference in the release behavior of cyclamate and orange peel essential oil in the commercial areca nut.

## 4. Conclusions

The present study investigated the compositions, texture characteristics and flavor release behaviors of four areca nuts. The main findings could be concluded as follows:

Among those four samples, the shape of AN1 was slender. AN1 also had higher soluble fiber and lower lignin, which were the basis of its lower hardness. Meanwhile, the TSS of AN1 was the highest, which indicated that AN1 had a moister and more succulent mouthfeel.

After the drying process, the degree of AN1 lignification was the lowest.

The commercially dried areca nuts CAN1 and CAN2 had the lowest hardness; meanwhile, the hardness of AN1 was relatively low compared to other homebrew dried areca nuts. AN1, CAN1 and CAN2 had higher alkaline pectin content and viscosity and better flavor retention, which could lead to better edible quality.

## Figures and Tables

**Figure 1 foods-12-01749-f001:**
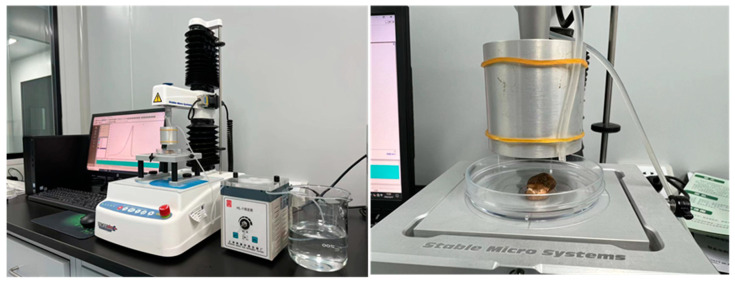
Flavor release experimental set-up.

**Figure 2 foods-12-01749-f002:**
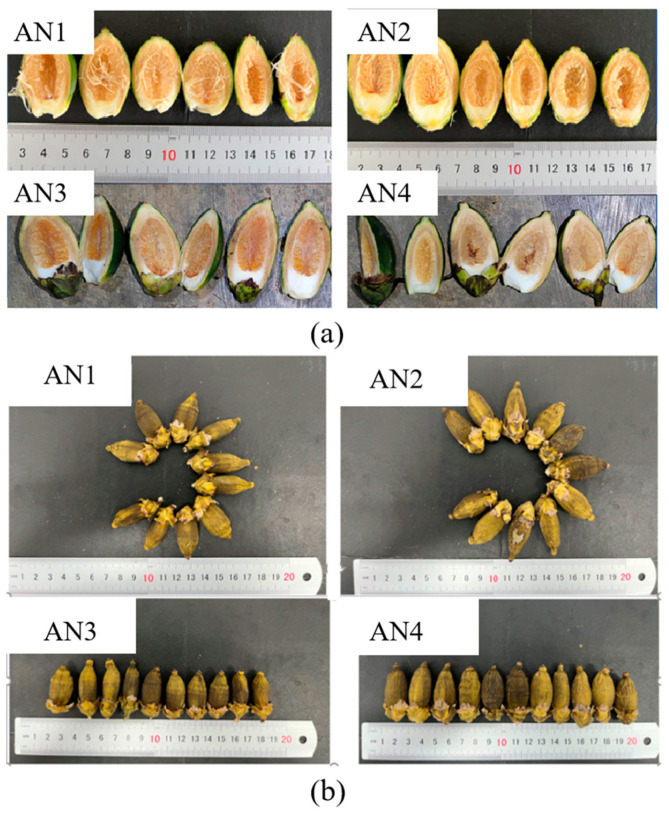
Visual appearance of areca nuts: (**a**) fresh areca nuts; (**b**) dried areca nuts.

**Figure 3 foods-12-01749-f003:**
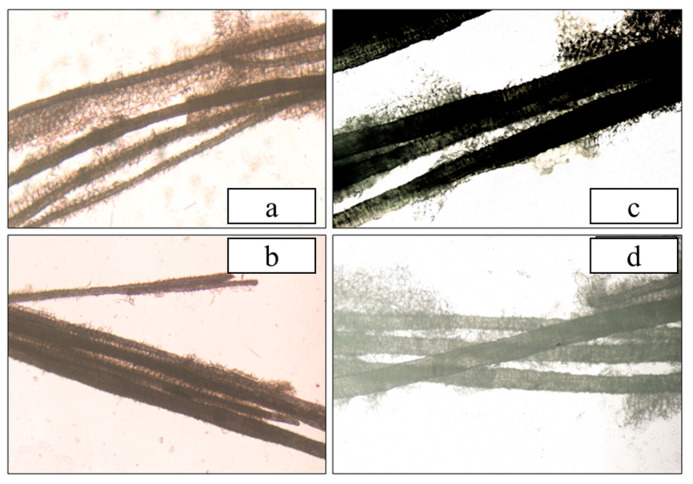
Microstructure of areca nut fibers: (**a**) AN1; (**b**) AN2; (**c**) AN3; (**d**) AN4.

**Figure 4 foods-12-01749-f004:**
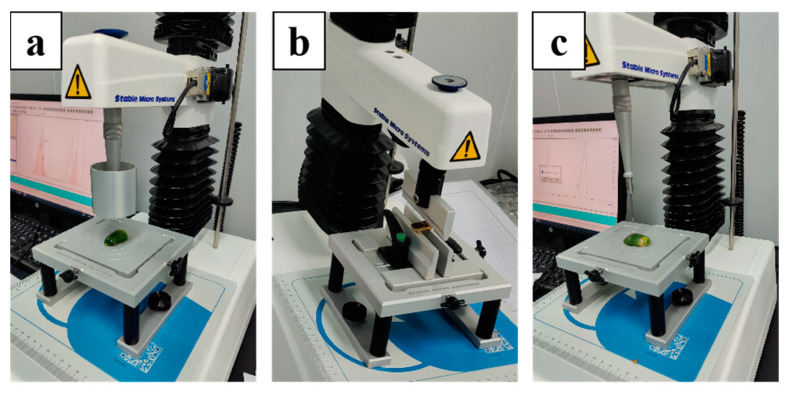
Texture analyzer with different probes: (**a**) compression test-P50 probe; (**b**) puncture test-P2N probe; (**c**) three-point bend test-HDP-3PB probe.

**Figure 5 foods-12-01749-f005:**
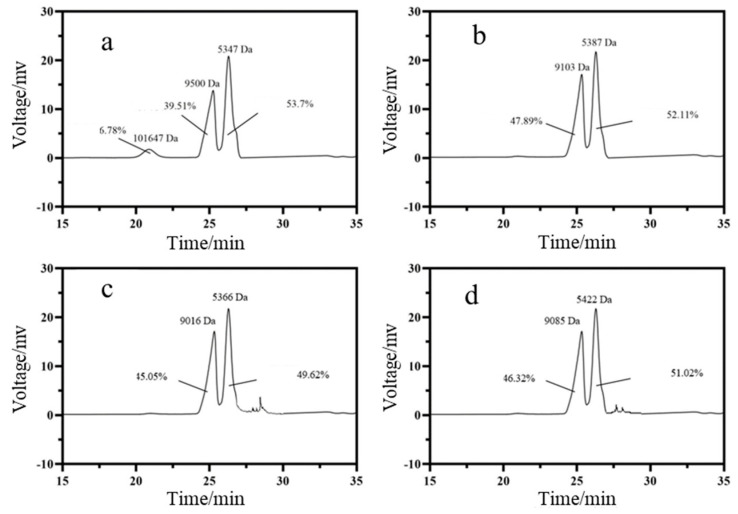
High−performance size exclusion chromatograph of areca nuts’ leachability: (**a**):AN1; (**b**) AN2; (**c**) AN3; (**d**) AN4.

**Figure 6 foods-12-01749-f006:**
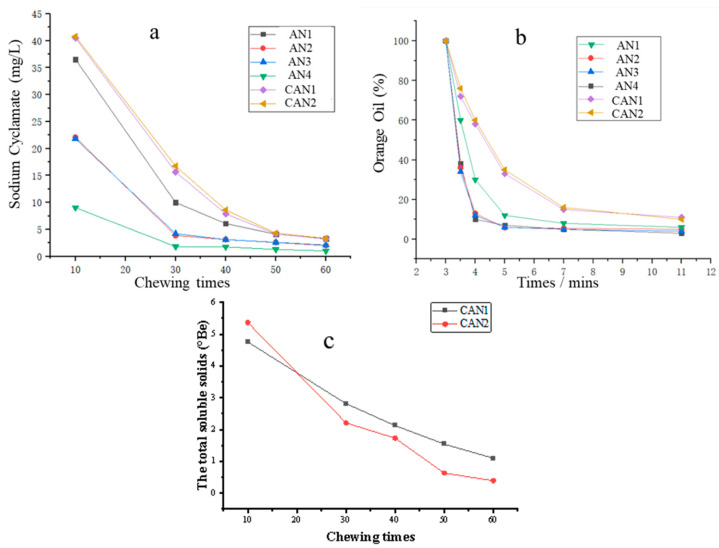
Flavor release behavior of areca nuts: (**a**) release behavior of sodium cyclamate; (**b**) release behavior of orange oil; (**c**) release behavior of soluble solids.

**Table 1 foods-12-01749-t001:** Dimensions of fresh areca nuts.

Areca Nuts	Length (mm)	Width (mm)	Length/Width
AN1	41.0 ± 3.1a	24.3 ± 1.1a	1.69
AN2	42.1 ± 3.9a	25.9 ± 3.5a	1.63
AN3	43.6 ± 4.1a	25.3 ± 3.3a	1.80
AN4	42.2 ± 3.1a	25.7 ± 3.4a	1.64

Values in the same column with different letters are significantly different (*p* < 0.05).

**Table 2 foods-12-01749-t002:** Composition of fresh areca nuts.

	TSS (%)	Soluble Fiber (%)	Total Polyphenol (mg GAE/mL)	Lignin (%)	Total Fiber (%)	Moisture Content (%)
AN1	4.94 ± 0.12a	3.00 ± 0.24a	2.76 ± 0.03b	24.81 ± 1.80b	26.07 ± 0.27a	78.26 ± 0.42a
AN2	4.73 ± 0.18ab	2.92 ± 0.25a	3.27 ± 0.02a	25.77 ± 1.74a	26.71 ± 0.52a	78.16 ± 1.27a
AN3	4.68 ± 0.15b	2.77 ± 0.28a	3.28 ± 0.05a	28.31 ± 1.86a	26.75 ± 0.62a	77.32 ± 1.52a
AN4	4.72 ± 0.10ab	2.86 ± 0.23a	3.31 ± 0.03a	26.01 ± 1.92a	26.16 ± 0.52a	77.74 ± 2.07a

Values in the same column with different letters are significantly different (*p* < 0.05).

**Table 3 foods-12-01749-t003:** Texture analysis of fresh areca nuts.

	Hardness (Puncture Test)/g	Hardness (Compression Test)/g	Chewiness (Compression Test)/g	Springiness (Compression Test)
AN1	345.96 ± 141.45b	23,775.91 ± 4502.57ab	2223.06 ± 686.69b	0.3038 ± 0.0253b
AN2	468.58 ± 158.65a	25,892.62 ± 4653.75a	3555.12 ± 698.38a	0.3128 ± 0.0368ab
AN3	508.50 ± 163.71a	28,270.76 ± 4777.90a	3703.98 ± 767.44a	0.3419 ± 0.0173a
AN4	498.74 ± 150.39a	26,266.14 ± 4654.20a	3595.29 ± 759.15a	0.3269 ± 0.0294ab

Values in the same column with different letters are significantly different (*p* < 0.05).

**Table 4 foods-12-01749-t004:** Fiber composition of dried areca nuts.

	Soluble Fiber/%	Lignin/%
AN1	1.94 ± 0.09a	26.07 ± 0.27c
AN2	1.62 ± 0.07b	27.25 ± 0.47b
AN3	1.53 ± 0.06b	28.71 ± 0.52a
AN4	1.58 ± 0.05b	27.57 ± 0.50b
CAN1	1.36 ± 0.03c	23.56 ± 0.33d
CAN2	1.31 ± 0.10c	24.65 ± 0.76d

Values in the same column with different letters are significantly different (*p* < 0.05).

**Table 5 foods-12-01749-t005:** Texture analysis of dried areca nuts.

	Hardness/g	Chewiness/g	Springiness
AN1	9396.36 ± 2184.87c	1422.44 ± 413.30a	0.3006 ± 0.0364a
AN2	10,452.56 ± 4251.33bc	1598.52 ± 455.38a	0.3119 ± 0.0382a
AN3	12,554.55 ± 3952.30a	1648.65 ± 513.32a	0.3165 ± 0.0389a
AN4	11,855.95 ± 3125.41ab	1552.34 ± 486.27a	0.3163 ± 0.0375a
CAN1	6069.00 ± 2537.69d	1025.61 ± 305.80b	0.2157 ± 0.0242b
CAN2	6064.66 ± 1626.23d	1035.29 ± 351.64b	0.2189 ± 0.0218b

Values in the same column with different letters are significantly different (*p* < 0.05).

**Table 6 foods-12-01749-t006:** Alkaline pectin content and viscosity of dried areca nuts.

	Content/g	Viscosity/Pa·s
AN1	4.66 ± 0.12a	0.0315 ± 0.0023a
AN2	4.15 ± 0.10bc	0.0205 ± 0.0018b
AN3	4.06 ± 0.10c	0.0215 ± 0.0021b
AN4	4.23 ± 0.15b	0.0203 ± 0.0027b
CAN1	4.57 ± 0.05a	0.0316 ± 0.0023a
CAN2	4.55 ± 0.06a	0.0317 ± 0.0023a

Values in the same column with different letters are significantly different (*p* < 0.05).

## Data Availability

The data presented in this study are available on request from the corresponding author.

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
