# Peer review of "Edible Quality Analysis of Different Areca Nuts: Compositions, Texture Characteristics and Flavor Release Behaviors"

_foods, 2023, doi:10.3390/foods12091749_

Round 1

Reviewer 1 Report

In general, the manuscript is well-written and contains interesting information for the readers. However, some comments should be considered and changed:

Line 57: If the authors purchased areca nuts from three different countries, how did they control for the initial condition of the nuts? Do they think the nuts are comparable?

Line 60: Please, could the authors specify the number of fruit used?.

Line 179: If I am not wrong, the AN3should be AN2.

Line 190: How can the authors demostrate the AN1 fiber was softer looking at figure 3?.

In tables 1 to 6, if the ANOVA test was significant, I would request the authors to carry out a post hoc analysis (tukey's test) and add the obtained results. This will make it easier for the reader to follow the results.

  •  

Reviewer 2 Report

This study compared the composition, texture characteristics and flavour release behaviours of areca nuts from different area and 2 commercial dried areca nuts. The authors evaluated characteristics of areca nuts before and after drying. Overall, this is an interesting, but small in scope study, showing possibilities for the development of areca nuts market. The authors need to consider the following comments.

 Abstract

Lines 17-18: “AN1, CAN1 and CAN2 have higher alkaline pectin content and viscosity, and better flavor retention, which indicated better edible quality “.  From the sentence written like this, it can be understood that nuts were characterized by viscosity.

Introduction

Line 50: What do you mean by edible quality?

Lines 53-54: The authors argue that the research information will promote harmless areca nuts consumption. Please, clarify which of the nuts characteristics allows you to make that assumption.

 Methods

 Line 91: Please, provide more detailed description of the method for the viscosity measurements. Was the shear rate constant during this experiment?

Line 162-163: Please, provide more detailed description of HPLC and GC analysis used for the cyclamate and limonene analysis. 

 Results and discussion

Why the authors did not provide data on the chemical composition of commercial nuts? From the discussion of the results, it is clear that they were high in sugars. Data on the amount of sugar and other substances in the commercial nuts would be helpful to understand the results obtained.

Chapter 3.8. Is it really only pectin that determines the viscosity of the solution? After all, other substances may be dissolved in it, which can affect the viscosity.

Why didn't the authors evaluate the aroma compounds that may be naturally present in nuts?

Chapter 3.9. I think it is incorrect to compare laboratory samples with commercial samples of nuts without providing the chemical composition of commercial samples.

 The figure 4 can be found in all textbooks on texture analysis. It does not provide any information that is discussed in the text. I suggest removing the picture

Round 2

Reviewer 1 Report

The manuscript now includes all the comments and revisions requested in the first revision. However, it still needs to be added in the statistical section of the methodology which method of mean separation has been used for all analyses. 

Reviewer 2 Report

 The manuscript has been sufficiently improved.
